# Antimicrobial Resistance Policy Protagonists and Processes—A Qualitative Study of Policy Advocacy and Implementation

**DOI:** 10.3390/antibiotics11101434

**Published:** 2022-10-18

**Authors:** Olivia S. K. Chan, Wendy W. T. Lam, Keiji Fukuda, Hein Min Tun, Norio Ohmagari, Jasper Littmann, Xu Dong Zhou, Yonghong Xiao, Ping Liu, Didier Wernli

**Affiliations:** 1The School of Public Health, Li Ka Shing Faculty of Medicine, The University of Hong Kong, 7 Sassoon Road, Pokfulam, Hong Kong SAR, China; 2National Center for Global Health and Medicine, 1 Chome-21-1 Toyama, Shinjuku City, Tokyo 162-8655, Japan; 3Division for Infection Control and Environmental Health, Norwegian Institute of Public Health, Skøyen, P.O. Box 222, N-0213 Oslo, Norway; 4The Institute of Social and Family Medicine, School of Medicine, Zhejiang University, 866 Yuhangtang Road, Xihu District, Hangzhou 310058, China; 5State Key Laboratory for Diagnosis & Treatment of Infectious Diseases, The First Affiliated Hospital, School of Medicine, Zhejiang University, Hangzhou 300013, China; 6Global Studies Institute, University of Geneva, Sciences II, Quai Ernest-Ansermet 30, CH-1211 Genève, Switzerland

**Keywords:** antimicrobial resistance, policy advocacy and implementation, protagonists, institutionalization, grounded theory framework

## Abstract

Antimicrobial resistance (AMR) fundamentally weakens societal foundations economically and in health care. The development of well-considered policies against AMR is important. However, in many places, AMR policy implementation remains elusive. This study aims to identify enablers and deterrents as well as processes and conditions in AMR policy advocacy. It also aims to identify AMR implementation conditions where AMR national policies are adopted and, to a certain extent, formulated and implemented. This study adopts qualitative research methodology and applies the Grounded Theory Framework to identify thematic findings from interviews conducted in China, Japan, Norway, the United Kingdom (UK), and the United States of America (US). It was identified that AMR policy protagonists are critical to filtering AMR issues and identifying policies “fit to prioritize” and “fit to implement”. They have helped move policy prioritization needles in the UK and the US and engaged in diplomatic efforts in the UK. In these cases, no clientelism was considered. In the US, protagonists who talked to the right decision-makers in the right office at the right time both moved AMR issues from individuals to institutional agenda and from social norms to policy agenda. **To conclude,** there are three thematic policy conditions that are significant to AMR policy advocacy and implementation: committed personal championship, institutionalization of policies, and social norms facilitate AMR policy advocacy and implementation.

## 1. Introduction

COVID-19 has compellingly made the case that certain threats of infectious diseases can not only harm individuals directly but undermine the functioning of countries. Similar to severe pandemics, antimicrobial resistance (AMR) has the potential to fundamentally weaken societal foundations [1,2]. Due to AMR, infections that once were reliably treatable are becoming more complicated or lethal. The pattern is global and is threatening the ability of countries to deliver modern health care and food security [3,4]. The development of well-considered policies against AMR is necessary to focus and guide resources, attention, and support. In 2015, WHO adopted a Global Action Plan on antimicrobial resistance [5,6]. More than 120 countries have elaborated a national action plan. However, in many locations, the implementation of AMR policies remains elusive [7,8,9]. 

In past years, countries have strived to formulate AMR policies based on evidence, data, and scientific consensus on AMR transmission pathways, contributing practices, and conditions [10,11,12]. However, the advocacy also heavily relies on the economic, political, and social relationships among health care systems, food production, trade, and domestic productivity costs [13,14]. By the same token, solutions to reduce AMR often require deeper investigations into infrastructure and governance problems with equitable access to universal health care, essential medicines, the use of antibiotic-free commodities, reduced antimicrobial prescriptions, and consumption [15,16]. For some policy administrations, AMR policy development has magnified the merit and weakness of governance, infrastructure, and policy instruments both nationally and sub-nationally [17,18]. What conditions and factors do protagonists and decision-makers consider that facilitate AMR policy advocacy?

To address the question, we conducted a qualitative study to identify some of the enablers and deterrents and the processes and conditions. These variables affect AMR policy agenda-setting and advocacy among selected countries where AMR national policies are adopted and, to a certain extent, administrated. The narrative is identified so other countries might use some of the lessons and advance their own policies. Informants are purposefully recruited based on their roles centering on antimicrobial resistance and their policy representation in public services, public health, human health care, food–animal agriculture, pharmaceuticals, and the environmental sectors. A total of 34 open-ended, semi-structured, in-depth interviews were conducted. Interview coding and thematic development were conducted according to grounded theory [19,20,21,22].

For clarity and comprehensiveness of the study, interviews were conducted based on a 30-item checklist that was established from the AMR-Intervene and AMR-PACT policy variables (Table 1) [23,24]. Interview analysis identified three main themes that characterized common and contrasting AMR policymaking. The findings facilitated a better understanding of how policies for AMR have been developed in China, Japan, Norway, the United Kingdom (UK), and the United States of America (US). Some of the themes helped identify strategies to address the challenges and to examine pathways and conditions that enable or deter AMR policy implementation in different contexts. In summary, it was identified that AMR policy protagonists are critical to filtering AMR issues and identifying policies “fit to prioritize” and “fit to implement”. They have helped move policy prioritization needles in the UK and the US, and engaged in diplomatic efforts in the UK. In these cases, no clientelism was considered. In the US, protagonists who talked to the right decision-makers in the right office at the right time moved AMR issues from individuals to institutional agenda and from social norm to policy agenda. Comparatively, the protagonists’ effects were invisible in political agenda-setting in China and Japan. It was opined that AMR experts in China and Japan tend to look to international peers for policy prioritization and political symbolic support. In all cases, unless AMR policies were institutionalized as exemplified in Norway, interviewees opined both public and private entities need to find ways to fund programs, institutionalize policies, and build social norms to sustain AMR policy in the long run.

## 2. Results

A total of 34 interviews were conducted from 2019 to 2020 (Table 2). Overall, we identified 97 codes that can be summarized as six themes (Appendix A). We organized themes into three main areas. First, individual championship is pivotal but insufficient in AMR policymaking. Observation centers on the importance of championships and where they are sub-optimal. Second, the institutionalization of AMR policymaking is key to sustaining and implementing AMR policies. Third, free markets play an ambivalent role in AMR policymaking while social norms can be an enabler in policy implementation.

Of all informants, 12 interviewees are policy advisors to national action plans or personally champion the AMR agenda at ministerial or diplomatic levels. Fifteen interviewees are policy advisors to sub-national policy implementation. Sixteen interviewees are technocrats on AMR steering committees, policy representatives in agriculture, and human health care professional bodies. Six interviewees are bureaucrats in AMR leadership roles who formulate or adopt AMR policies. A summary of grounded theory theme-based and country-based key findings is listed in Table 3.

### 2.1. Individual Championship Is Pivotal but Insufficient in the AMR Policymaking Arena

Individual champions were pivotal to advocate, initiate, and implement AMR policies. These AMR policy protagonists generated awareness, used bureaucratic ties, and lobbied decision-makers to raise the profile of AMR on the agenda based on personal, professional, public health, and social responsibilities. The success of these individual champions in shaping policy depended on access to key decision-makers in Norway, the US, and the UK. Policy advocacy on AMR was most successful when coupled with institutional and bureaucratic buy-in, bottom-up social norm support, and multistakeholder partnerships. By contrast, the lack of unified social norms on seeing antimicrobials as a common good gave a weaker voice to changing or prioritizing AMR on agendas in Japan and China. Some of these personal efforts had gone as far as directly changing national policy prioritization, overcoming diplomatic gridlock, and in some cases sustaining AMR policies in the long term. In Japan and China, AMR policy protagonists agreed that AMR policymaking would benefit from international support on technical and policy capacity building. In China, AMR policy personal champions were mostly technical advisors who have tried but were not able to directly influence key political decision making at national or sub-national levels. Alternatively, the absence might simply be due to the marked separation between the political and public health spheres 

All interviewees who were AMR policy protagonists found implementation of AMR policies challenging. Public health, medical, and policy experts who were able to influence policymakers indicated their frustration of failing to foresee or bridge the gap between prioritizing AMR policy at national levels and the implementation of policies at sub-national and sectoral levels. Interviewees from the UK, the US, Norway, China, and Japan recalled their actions during policy advocacy and that there was “little consideration” to see beyond the initial stage of the initiation of the policy process. Other UK policy experts emphasized anticipating conditions for policy implementation was just as important as policy initiation. The interviewees opined that the consequences of such gaps lead to failure to provide the needed infrastructure, resources, and governance for AMR policy implementation. Interview quotes are included as 2.1.1 to 2.1.4 in Appendix A.

AMR policy protagonists who were allowed to advise policy at stages of policy initiation and formulation helped converge public, professional, and policy perspectives. They introduced or recombined relevant information, especially in the case of policy initiation impasse [25]. Different societies, however, permitted protagonists to become involved in AMR policies at different stages of the policy process and with varying levels of influence. Compared to Norway, the UK, and the US where interviewees were involved in the early stages of the policy process, interviewees in China and Japan were recruited at a later stage, some for technical advice after the government agenda was set and decisions about priorities were made. Additionally, interviewees who worked to influence AMR policies in China and Japan opined there was lower accessibility to political decision making. This was opined partly due to accessibility problems to key decision-makers in a classical hierarchical governance approach. In addition, the data collected highlighted intrinsic differences among different policy situations. In the UK, the US, and Norway, interviewed policy protagonists were involved in discussion during policy prioritization and spoke directly to political decision-makers. The US interviewees also used leverage from society and the media when advocating AMR policies. Interviewees from China emphasized a more top-down approach to AMR policymaking Interview quotes are included as 2.1.5 to 2.1.9 in Appendix A. 

Comparatively, there were specific policymaking considerations within China and Japan. We found two distinct patterns in policy advocacy and implementation. First, interviewees from China and Japan—though some were considered by their countries as experts—described fewer links between technocrats and those that were politically powerful to influence policies. It was even less common for these individuals to have experience or access to political nexuses or to negotiate for diplomatic support. Overall, personal championship indicated a more important role in the UK and the US than in Japan or China. It appeared that it was uncommon culturally and socially in China or Japan to confer an important advisory role to an individual outside of established political structures. Most individuals interviewed in China and Japan had tried to reach or influence AMR policy decision making. Most interviewees from these countries also described a lack of political interest and political commitment as one of the major obstacles to the implementation of AMR policies. Interviewees from China and Japan opined that international peers were an important source of leadership to advocate and implement AMR policies.

### 2.2. Policy Institutionalization Facilitates AMR Policy Prioritization and Implementation

Personal championship was an essential enabler in AMR policy advocacy and also for maintaining long-term policy commitment. In some cases, reduced involvement from an individual champion was associated with a loss of AMR policy continuity. In cases when personal champions played a key role in the adoption of AMR policies by bureaucracies without institutionalization, lack of institutional memory was prominent in some countries. This lack of institutional memory continued to revert AMR policies to their earlier stages. For instance, during the discussion of AMR policy implementation in China, interviewees observed that individuals who led the AMR work often engage in endogenous effort and resources to influence guidelines and stewardship programs at the implementation phase and settings. However, the implementation and policy fidelity were individual-dependent and therefore more likely to be inconsistent. In the US, a change of political leadership led to AMR policy deprioritization. This emphasized the issue of long-term commitment to addressing AMR. Interview quotes are included as 2.2.1 to 2.2.3 in Appendix A. 

Interviewees suggested that prioritization and resource commitment on AMR need to transition from personal advocacy to institutionalization. Personal advocacy was much more prominent in the UK and the US than in Japan and China. Among these four countries, a commonality underpinned the ultimate implementation of policies—an individual or individuals from hospitals, farming, community pharmacies, public service systems, medical systems, and government offices stepped up endogenously, or in other cases followed through with their jobs. Interviewees opined AMR governance around organizational collaboration, orchestration, establishment, and enforcement of policies, guidelines, and regulations was built on personal investment and motivation rather than well-defined institutional processes. These findings stood in contrast with findings from Norway, which were indicated to be incorporated and institutionalized AMR policies based on their existing governance and administration capacity rather than by relying on individual champions.

The example of Norway illustrated some of the factors that were important in supporting countries’ effective capacity to address AMR. Interviewees described sustained cross-sectoral collaboration to address AMR, which was based on a mutual understanding of its significance for local and global health and could use well-established infrastructure and political instruments. In Norway, AMR control, according to experts from both the public health sector and the veterinary sector, was seen as a core part of infectious disease control, food safety, and zoonotic disease prevention. Well-established cooperation among human, animal, and food sectors at government and non-government offices also helped AMR policy development. As a result, the governance, infrastructure, and cross-sectoral relationships provided the backbone for AMR policies. By comparison, interviews in the US, the UK, Japan, and China indicated a lower level of support for collaborative governance and policy prioritization to institutionalize which usually takes time and resources.

### 2.3. Free Markets Play an Ambivalent Role While Social Norms Are a Driver in AMR Policymaking

Protagonists opined that economic and financial incentivization to increase antimicrobial innovation and reduce antimicrobial use was still very much in the hands of private industries. For antibiotic innovations, there were non-profit mechanisms such as Combating Antibiotic Resistant Bacteria Biopharmaceutical Accelerator (CARB-X) and Global Antibiotic Research & Development Partnership (GARDP) which brought new antibiotics to market [26]. However, interviewees in the UK and China who worked closely with pharmaceutical development and medical insurance policies emphasized the need for a “business case” to sustain AMR innovation policies. Interviewees also opined that there were not sufficient public–private partnerships, especially in the face of hurdles in drug legislation, cost of drug development, and low potential for financial profit. For reducing antimicrobial use, the US interviewees stated contemporary public and private medical insurance reimbursement mechanisms that encouraged patients to use medical facilities and services with infectious disease prevention and control programs had also facilitated antimicrobial use stewardship programs. Additionally, in China, financial disincentives changed antimicrobial prescription and consumption behavior in hospitals [27].

Without social norms, interviewees opined that financial direction among antimicrobial users ultimately overrode the protagonists’ influences—be that antimicrobial stewardship, antimicrobial production quality control, or pharmaceutical innovation. Interviewees emphasized that local social norms such as patient antibiotics use compliance, food-producing animal farming industrial practices, and consumers’ willingness to pay for antibiotics-free food influenced antimicrobial use. For instance, interviews from Norway confirmed strong social values and norms to protect public health as a key determinant behind AMR policy advocacy. Additionally, the younger generation was perceived to be the next social norm to facilitate prudent antimicrobial use in China. On the contrary, an interview in Japan provided insight that perceived hurt to national economics has driven social norm and policymakers to side with softer or different policies. Interview quotes are included as 2.3.1 to 2.3.7 in Appendix A. 

### 2.4. Summary of AMR Policy Advocacy and Implementation Variables

Variables that influence policy protagonists and AMR policy advocacy and implementation can be categorized as three levels of micro-, meso-, and macro-determinants (Figure 1). Personal championship represented by personal belief, professional obligation in medicine and public health are represented as micro-determinants. The political, social, cultural, and economic variables influencing policy protagonists and their actions in advocacy and implementation are represented as macro-determinants. The organizational dynamics and conditions such as an organization’s memory of policies, institutional management coherence, governance, and infrastructure capacity are represented as meso-determinants. These three determinants and their relationships either enabled or inhibited policy protagonists in exercising their advocacy roles and capacities, institutionalization of policies for policy sustainability, and policy climate for advocacy or implementation of AMR policies. 

## 3. Discussion

We identified commonalities and variations in policymaking that are pertinent to the design and implementation of effective and sustainable AMR policies. First, AMR policy requires resources and planning, especially for implementation, which has proved to be a challenge in many countries. Though initiation and advocacy of policies were described, most countries either did not factor in or found out that they underestimated the amount of resources needed to successfully implement the policies in their action plans. Second, personal championship works well to advocate AMR policies in the respective cultures and societies of the US and the UK. However, individual champions from China and Japan do not appear to have equal success in advocating for policy change. Rather, Japan and China’s AMR policy advocacy benefits from strong international leadership and symbolic policy advocacy. Third, AMR policymaking is challenging in a public health network and social norm which are not ready to pull together politically and economically sustainable AMR efforts. Rather than relying solely on the private market for antibiotics innovation or stewardship, public–private partnerships and social norms are needed to overcome economic systems that are less developed in considering AMR as a public health entity.

Policies for AMR are still at an early level of development in some countries for several reasons—in countries where individual championship is viewed with skepticism, policy development and implementation have often stalled and are not yet incorporated into bureaucratic structures and processes. For policy advocacy that relies on international and national peer influences, policies drafted and programmed were not given sufficient in situ consideration to tailor to sub-national and sector contexts. For policies that landed without infrastructure and governance, AMR advocacy was perceived to be a waste of political credit or time.

Our article had several limitations. The first was the difficulty to recruit policy protagonists and decision-makers, especially in China and Japan. The narrative from decision makers in AMR policy development remained limited in these two countries. The second limitation was to recruit sufficient representation in face of the separation of technical protagonists and political decision-makers. The third limitation was interview variation that could occur with different interviewers’ office positions in different countries. Despite these limitations, three key points can be considered in some countries around AMR policy advocacy. First, public health offices should involve leaders in the field to ensure existing policies are being effectively implemented. Policymakers should engage these technical and policy protagonists to answer AMR policy questions because protagonists can help share the burden of policy decisions, fine-tune policy drafting, and help campaign for AMR policy implementation. Second, public offices can focus on policymaking at governance and infrastructure building, transitioning protagonists’ recommendations to sub-national levels, and focusing on institutionalizing AMR policymaking. Third, social norms are an important variable for policy advocacy and resource commitment in policy implementation. 

## 4. Methods and Material

### 4.1. Sampling Technique, Enrolment, and Interview Methodology 

Countries were included for their diverse contexts, different policy advocacy processes, and policy processes around AMR national policies. Interviewers were key collaborators (technical working group, TWG) who were invited to participate in this study because of their experiences and work in antimicrobial resistance policies. Training and interview guidance was provided to all interviewers. Interviews were piloted in Norway and China. Interviews were conducted in China (Mainland, Macau, and Hong Kong) (September 2018 to November 2019), Norway (March to May, 2019), Japan (June to August, 2019), the United Kingdom (September, 2019), and the United States (October, 2019 to February, 2020). 

Interviewees were individuals with track records of influencing AMR policies in China, Japan, Norway, the UK, and the US. Interviewees were primarily selected by knowledge of individuals’ expertise and their influences on AMR policies (Table 2). Further recruitment of interviewees was carried out through interviewees’ recommendations on local policy protagonists. 

For interview planning and clarity, the core questions list and policy variables were developed from the AMR-IMPACT and AMR-Intervene frameworks (Table 1). Face-to-face, semi-structured, open-ended interviews were conducted. All in-person interviews were audio-recorded and online interviews were audio- and video-recorded. For interviews conducted in a language other than English, transcripts were translated to English by OC, LP, and the TWG and validated by reverse translation and local collaborators. Interviews were continued until the themes of the interview content were saturated and when the content of interviews covered the different sectors that are relevant to AMR including food/animal production, food production, and human medical sectors in each country.

### 4.2. Data Analysis

All audio files were transcribed verbatim to Word documents. Familiarization with data was conducted by researchers listening to audio recordings (O.S.K.C. and P.L.) and reading interview transcripts (O.S.K.C. and P.L.). Transcribed interviews were imported into qualitative analysis software ATLAS-ti 8.4.25 (ATLAS ti Scientific Software Development GmbH. version 8) for analysis and codebook development. All identities of informants were delinked from the transcripts from the coding stage onwards.

Researchers read recursively and iteratively among data, codes, and themes for constant comparison. All codes were sets of pre-designated English letters and Roman numerals simplified from variables designated in AMR-PACT and AMR-Intervene [23,24]. All open codes were first assigned to endogenous and exogenous determinants that affected these policy protagonists [28,29,30]. Further open-, axial-, and selective coding was performed according to the Grounded Theory (GT) framework. Thematic and construct analyses were derived inductively according to selective codes [31,32,33,34,35]. The thematic reporting framework followed the Consolidated Criteria for Reporting Qualitative Research (COREQ) [36].

## 5. Conclusions

Though individual championships, or “principal coalition actors” according to the Advocacy Coalition Framework [37,38], are pivotal to establishing AMR policies, they alone are insufficient to sustain policy implementation. Additional policy processes, political will, and economic resources are needed to institutionalize AMR policymaking so that stakeholders and agencies can sustain and implement AMR policies. It is also observed that free markets play an ambivalent role in AMR policymaking while social norms are important to enable AMR policy implementation. Ethnographically, the protagonists’ effects were felt to be invisible in political agenda-setting in China and Japan. To that, informants opined that AMR experts in China and Japan tend to look to international peers for policy prioritization and political symbolic support. In all cases, unless AMR policies were institutionalized as exemplified in Norway, interviewees opined that both public and private entities need to find ways to fund programs, institutionalize policies, and build social norms to sustain AMR policy in the long run.

## Figures and Tables

**Figure 1 antibiotics-11-01434-f001:**
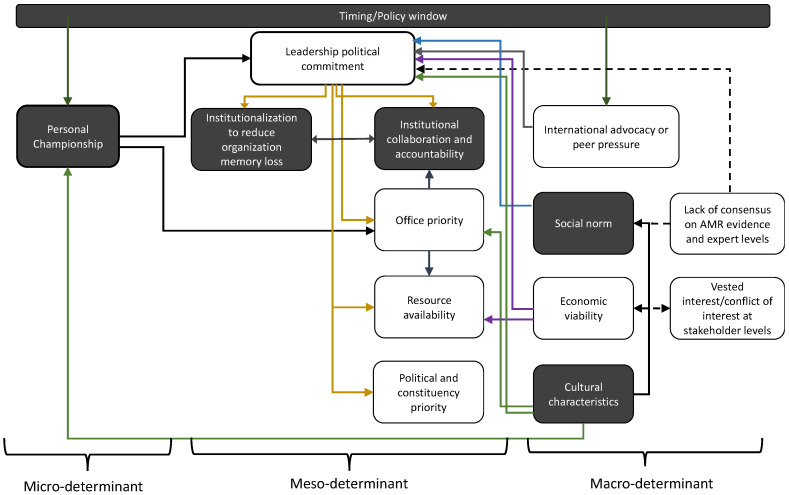
AMR policy protagonists consider multiple variables in their policy advocacy and implementation process. The main themes identified are indicated in black boxes. Solid arrows depict how variables are opined to have facilitated advocacy and/implementation. Dashed arrows depict deterrents in policy development. Yellow arrows depict political, green depict cultural, purple depict economic, and blue depict social characteristics. Micro-, meso-, and macro-determinants are bracketed.

**Table 1 antibiotics-11-01434-t001:** Core interview variables and sample questions categorized as micro-, meso- and macro-determinants.

	Index-Code	Numbers	Variables to Be Determined	Sample Questions		Index-Code	Numbers	Variables to Be Determined	Sample Questions
**Micro-determinants**	MI	1	Perceived responsibility in personal advocacy/personal championship	Why do you decide to advocate for AMR issues?	**Macro-determinants**	MA	1	Social influence and consumer behavior influences AMR policy advocacy	When it comes to civil society and social norm, can you say a bit more about that?
	MI	2	Perceived personal political capacity and connections	What was your role or goal in the (office/parliament/organization) related to AMR policies?		MA	2	International organization influence AMR policy advocacy	You can take an economic argument, you can take a political argument that everybody else is moving in this direction and will be left behind and so on. So, in terms of tools and strategies at that level were there any particular approach that (the country/office/organization) took?
	MI	3	Perceived risk or benefits in political career	Why did you decide that you needed to achieve to prioritize or put AMR on the agenda? How does that affect your role and responsibility in your (cabinet/office/instiution)?		MA	3	Economic influence or pressure influence AMR policy advocacy	
	MI	4	Perception of sufficient support to advocate, prioritize or formulate policy	Do you see that AMR and (the country) moving on AMR the stars aligned and a group of things came together. What were the factors that prompt you to push for AMR?		MA	4	Resources and capacity availability limits or enables AMR policy advocacy	What logic, what arguments, what other factors did you find most important in successfully taking AMR forward? Including with peers and others. Specifically, how did you use health, science, politics, economic perspective
	MI	5	Of opinion that AMR is a traceable problem translatable to policy	….question then is, what were the primary goals that you were calling when taking on an AMR? What do you see about AMR that means to you in terms of advocating for prioritization on the agenda?		MA	5	Legislation ease, hurdles, and poltical climate influences AMR policy advocacy	Do you think that this was an approach primarily as a health issue that the political side could help facilitate, or, it was really more of a political issue and this was an opportunity to move on that in terms of, moving ahead with (country/s) concerns and enrolling others and trying to further those political aims?
	MI	6	Perception of positive possibility to prioritize AMR on agenda	What were the most important reasons for you or for your Department in deciding to make AMR a major priority?		MA	6	Collaboration or isolation among different offices influences AMR policy development	What was the step, or what would have been the step for translating that Department of Health document into a priority, which mobilized you in the (your) Office?
	MI	7	Of opinion that AMR is a solvable issue	Do you think AMR is a solvable issue?		MA	7	Consensus/social norm among entities and organization within country influences AMR policy development	Were they driving it? Were they holding it back? Were they as important as the government voices in initiating this conversation?
	MI	8	Of opinion that AMR aligns with decision maker’s core belief	Do you think when you spoke to (Decision maker/policy entrepreneur), do you think AMR strikes with his/her core belief?		MA	8	Global and regional collaboration influences AMR policy advocacy	In the beginning when you were looking at AMR both as a national issue and as you get into the complexity, you see it more as an international issue. How did the perspective or goals evolve? Did they become more specific or general?
	MI	9	Of opinion that AMR is a priority within office or former office	What was your job title and what were your general responsibilities? In the context of that, can you say a little bit about how your working relationship works in the day-to-day basis of your posting? How and what you would say about your office’s role in AMR policies(Personal/professional responsibility clarification)?		MA	9	Professional support or not influences AMR policy advocacy	Do you think AMR policy has the traction to win over everyone (medical professionals, farming industries, pharmaceuticals) that had to be won over? Do these stakeholders do what they have to do and is that important?
	MI	10	Previous knowledge and experience influence belief in mitigating AMR			MA	10	Cultural, historical, geographical and anthropological contextual influences AMR policy development	Could these have come together for another topic or was AMR a relatively unique among the different health issues that, you know, (the country) was facing? Could (your country’s) cultural, historical, geographical and anthrological context influence how AMR policy developed?
Meso-determinants sample questions:									
In the context of tackling AMR, do your office/position require collaboration with other entities? What is the level of collaboration? (Intermittent versus consistent communication, intermittent versus consistent knowledge exchange, uni-sector or complete program integration) What are some of the factors that makes your collaboration with…( )…particularly successful? What worked or what did not work? How did you make the collaboration possible? What was done to resolve issues in prioritizing/adopting/implementing AMR policy in and around your institution/agency/organization/profession? Is AMR a priority in your office’s agenda? Why is it a priority/not a priority? What/Who influences your decision/advice? Does AMR strike your organization’s core value or belief? What does it If so, what are the drivers to your AMR policy advice/decision? Does your organization see AMR issue as something that can be curbed/tackled? Does your organization see a timeline to advocate/adopt/implement AMR policy/solve the AMR issue? Does your organization/institution/profession think current AMR scientific evidence is sufficient? Is that important to the decision to advocate/adopt/implement AMR policies? Do you think external pressure such as consumer interest/public health/peer groups/country mandate/social norm affect your prganization’s decision/advice? Do you think international, offices or international mandate affects your organization/profession decision or advice? What is the cost of action vs inaction? Can you afford to shift the allocation of resources towards AMR or away from AMR policies? Is resource allocation and management an important factor for your organization to advocate/adopt/implement AMR polciies?									

**Table 2 antibiotics-11-01434-t002:** Summary of interviewee profiles.

		Main Position(s) and Role(s) Associated with Antimicrobial Resistance Policymaking					
Countries	Number of Interviewees	Policy Makers	Hospital Directors and Leaders	Healthcare and Public Health Officials	Medical Professionals (Pharmacists, Dentists, Doctors & Veterinarians)	Diplomat, Bureacrats, and Policy Administrators	Policy Advisors
**China** *	12	3	2	3	10	3	8
**Japan**	5	0	3	1	3	0	3
**Norway**	6	1	0	2	2	0	5
**UK**	6	2	0	1	2	2	6
**USA**	5	1	0	3	4	1	5

* including Mainland, Hong Kong, and Macau.

**Table 3 antibiotics-11-01434-t003:** Grounded Theory thematic and country-based key findings.

	Thematic Key Findings	Country-Based Key Findings
**1**	Personal championship is key to initiation of AMR policy advocacy.	Norway’s AMR policy protagonists viewed unified understanding against AMR enables its policy advocacy. Interviewees also see coherent public health, healthcare, and farming system facilitators of AMR policy implementation.
**2**	Timing is important for policy protagonists to advocate and/or implement AMR policies	The UK AMR protagonists mobilized national effort to influence international and One Health AMR advocacy.
**3**	Institutional memory is important for policy durability and especially for implementation, which has proven to be a challenge in many countries.	China AMR policy protagonists sees top-to-bottom policies that advocate and implement AMR policies in food security, professional training, public education, and antimicrobial regulation issues.
**4**	Institutionalization, institutions’ connectedness, and collaboration are keys to sustain policy development but still faces challenges in change of constituencies.	The US policy protagonists view public-private partnership in AMR a key approach in policy advocacy and implementation in hospital and health insurance sectors.
**5**	Free market plays an ambivalent role while social norm is important enabler in AMR policy prioritization.	Japan and China’s AMR policy advocacy benefit from strong international AMR leadership and policy advocacy.
**6**	Culture, socioeconomic, and ethnographic variables enable or inhibit policy protagonists in AMR policy advocacy and/or implementation

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
