# Peer review of "Antimicrobial Resistance Policy Protagonists and Processes—A Qualitative Study of Policy Advocacy and Implementation"

_antibiotics, 2022, doi:10.3390/antibiotics11101434_

Round 1

Reviewer 1 Report

Thank you for this opportunity of reviewing this manuscript. This qualitative study conducted interviews with people who have influenced AMR policies to identify similarities and differences among countries.

The successes and failures of the project revealed the current challenges, and the key points of "involve leaders in the field and policymaking at governance and infrastructure," "building "technical and policy protagonists" and "social norms.

The results is important information to understand how AMR policies have been developed in some countries and to facilitate future AMR policies.

The Types of Publications section of the Submission Rules states.  Wouldn't this text fall under Perspectives rather than Articles?

  • Perspectives: Perspectives are opinion or commentary articles that express personal opinions about existing studies that have great impact on the antibiotics research community. Perspectives should have a main text of around 2000 words at minimum, with at least 20 references. The same requirements apply to communications, brief reports, and other similar publication types.

Major comments:

<Methods>

·         Please add a table summarizing interviewees' nationalities, characteristics, positions, etc.

·         This study is semi-structured interviews, please add the core questions list and policy variables as a table or supplementary table.

·         Please add a corresponding table of coding done in this study.

<Results>

  • This paper is difficult to understand the main points because it is written only in text. We would like the results to include a table summarizing the main points of the survey results for each country.

<Discussion>

·         Is there any impact of the timing of policy formulation in each country and the timing of the next policy formulation (next AMR action plan) and interview implementation, although they are different? (e.g., the period during which the effects of the implementation of the formulated policy will spill over).

Minor comments:

<Result>

l  P. 2, In the Result section, it states that interviews were conducted until 2021, but in methods section until 2020. Which is correct?

l  P. 2, Please consider including the 97 codes and six themes identified, e.g. in supplementary materials.

l  Is it reasonable to assume representativeness of the Asian, European and North American regions in interviews with five countries? Please consider.

<Discussion>

l  The nationalities of interviewees seemed to be biased towards some, please consider including them in the limitation.

Author Response

Dear Reviewers, 

We really appreciate your comments and advices. We have included the changes in manuscript and we have also prepared a document with responses to comments. Please find attached file with point by point responses to comments. 

Reviewer 2 Report

It would be useful to see the 6 categories what were compiled fram the 97 codes!

Author Response

Dear Reviewers, 

We really appreciate your comment and input to make this manuscript better. 

Please find point to point response to comment in attached file. 

Reviewer 3 Report

1. The idea and purpose were quite good

2. The interview were already done, however, is it possible to look back and change the answer and characteristics of participant institute to be numbers or score that the authors could analyzed and show the scientific evidence better than just descriptions.

Author Response

Dear Reviewers, 

We really appreciate your comments and input. We wish to thank you for your comments that helps improve the manuscript. We have included point-by-point responses to comments in attached file. Changes are also made in the manuscript. 

Round 2

Reviewer 1 Report

Thank you for the correction to the appropriate comment. I will make one minor comment.

As mentioned in the first limitation, I would like to add the possibility of bias due to the difference in the position of the interviewer in each country.

Author Response

Dear reviewers, 

Appreciate your follow up comment. Please find supplementary table edited in manuscript document and responses in word document.

Reviewer 3 Report

Appreciated your hard work for the major revision, however, the manuscript is too long too read to understand the content and context. Could you please make it more concise.
Many things can move to appendix or supplement, especially the phrases from conversation of the interviews.

Author Response

Dear Reviewers, 

Appreciate your comments. Please find response in word document and supplementary table added to manuscript. 
